# KSHV infection of B cells primes protective T cell responses in humanized mice

Nicole Caduff[1,4,7], Lisa Rieble [1,7], Michelle Böni[1], Donal McHugh [1,5], Romin Roshan [2], Wendell Miley[2], Nazzarena Labo[2], Sumanta Barman [3], Matthew Trivett[2], Douwe M. T. Bosma [1,6], Julia Rühl[1], Norbert Goebels[3], Denise Whitby [2] & Christian Münz [1] ✉

Kaposi sarcoma associated herpesvirus (KSHV) is associated with around 1% of all human tumors, including the B cell malignancy primary effusion lymphoma (PEL), in which co-infection with the Epstein Barr virus (EBV) can almost always be found in malignant cells. Here, we demonstrate that KSHV/EBV co-infection of mice with reconstituted human immune systems (humanized mice) leads to IgM responses against both latent and lytic KSHV antigens, and expansion of central and effector memory CD4+ and CD8+ T cells. Among these, KSHV/EBV dual-infection allows for the priming of CD8+ T cells that are specific for the lytic KSHV antigen K6 and able to kill KSHV/EBV infected B cells. This suggests that K6 may represent a vaccine antigen for the control of KSHV and its associated pathologies in high seroprevalence regions, such as Sub-Saharan Africa.

Kaposi sarcoma associated herpesvirus (KSHV, HHV-8) and Epstein-Barr virus (EBV, HHV-4) are oncogenic γ-herpesviruses that infect large proportions of the human population and contribute an estimated 1-2% to the worldwide tumor burden[1,2]. γ-herpesvirus-associated lymphomas include endemic Burkitt's, Hodgkin's and diffuse large B cell lymphoma (BL, HL, DLBCL) for EBV, and plasmablastic multicentric Castleman's disease (MCD) for KSHV[3–7]. Furthermore, KSHV associated primary effusion lymphoma (PEL) is per definition KSHV positive and in about 90% carries EBV in the very same tumor cells[8,9].

Many of these malignancies develop primarily in the setting of a compromised adaptive immune system. Transplant recipients frequently develop Kaposi sarcoma (KS) and EBV-associated post-transplant lymphoproliferative disease (PTLD)[10] as a consequence of the immunosuppressive drugs that target T cell function to prevent allograft rejection[11]. Reduction of immunosuppression often improves clinical symptoms, and EBV+-PTLD can additionally be treated by adoptive transfer of EBV-specific T cells, thus restoring EBV-specific immune control[12,13]. Individuals with uncontrolled human

immunodeficiency virus (HIV) infection often succumb to EBV- and KSHV-associated malignancies due to their progressive immunosuppression[13]. The implementation of antiretroviral treatment against HIV has led to a marked decline of most mentioned malignancies, including KS, once the most common AIDS-associated cancer[14]. Yet, the incidence of EBV+ HL, PEL and MCD seems unaltered or increased despite effective HIV suppression[15–17]. The severe and often fatal outcomes of γ-herpesvirus infection in people with primary immunodeficiencies further underline the importance of robust immune surveillance of EBV and KSHV. Although the underlying monogenetic alterations that predispose for uncontrolled infections and their associated pathologies differ between EBV and KSHV, they often compromise T cell mediated immune control[18]. While no association between genetic immunodeficiencies and increased frequencies of KSHV+ B cell malignancies (i.e. PEL, MCD) have been identified so far, mutations affecting T cell receptor (TCR) signaling (STIM1, WASP), co-stimulation (OX40, MAGT1) and Th1 effector function (IFNγR1, STAT4) seem to increase the risk of KS, as reviewed in[18].

[1]Viral Immunobiology, Institute of Experimental Immunology, University of Zürich, Zürich, Switzerland. [2]Viral Oncology Section, AIDS and Cancer Virus Program, Frederick National Laboratory for Cancer Research, Frederick, MD, USA. [3]Department of Neurology, Medical Faculty and University Hospital Düsseldorf, Heinrich Heine University Düsseldorf, Düsseldorf, Germany. [4]Present address: Genentech Inc., 1 DNA Way, South San Francisco, CA 94080, USA. [5]Present address: Pfizer, Medical Department, Schärenmoosstrasse 99, 8052 Zürich, Switzerland. [6]Present address: Department of Immunology, Leiden University Medical Center, Leiden, Netherlands. [7]These authors contributed equally: Nicole Caduff, Lisa Rieble. ✉e-mail: christian.muenz@uzh.ch

Similarly, primary immunodeficiencies that predispose for EBV-associated lymphoproliferations include deficiencies in TCR signaling (e.g. ITK, PI3K, RasGRP1, ZAP70), co-stimulation (e.g. CD27, CD70, CTLA-4, 4-1BB) and cytotoxic effector functions (Munc13-4, Munc18-2 and perforin)[18,19]. The importance of T lymphocytes, especially that of CD8[+] T cells, in controlling EBV infection has also been demonstrated in preclinical experimental models by CD8 depletion[20–22] or immuno-suppression via small molecules[23]. T cell targeted interventions of the human immune system in these in vivo models led to higher EBV viral burden and more frequent growth of EBV[+] lymphoproliferations, while the adoptive transfer of EBV-specific T cells transiently controlled high viremia[24]. Previous studies reported that purified T cells from KSHV-seropositive individuals respond to a limited number of KSHV gene products ex vivo[25] possibly favoring early lytic proteins[26], and that humoral responses against KSHV are highly variable between virus carriers[25,27]. In general, the properties and antigenic repertoire of adaptive responses against KSHV remain insufficiently explored, in particular in the context of KSHV-driven B cell lymphomagenesis.

Mice with reconstituted human immune systems (humanized mice) are susceptible to infection with KSHV[28,29], particularly when co-infected with EBV, and develop PEL-like KSHV/EBV B cell tumors[29]. This is in line with in vitro infection studies and a cohort study with African children, suggesting concurrent infection with EBV as a prerequisite for initial KSHV persistence in B cells[27,30]. The ability of humanized mice to exert antigen-specific control of KSHV infection has not been investigated so far.

Here, we show that KSHV and EBV co-infected humanized mice present with an increased number of cytotoxic T cells displaying an effector and central memory phenotype. We isolated KSHV-reactive T cells and show that they react specifically against the KSHV K6 protein and kill KSHV-infected B cells. This identifies a protective T cell specificity after KSHV infection of B cells in humanized mice that could be explored for vaccine development or immune therapies.

## Results

### KSHV infection elicits an expansion of CD8[+] T cells in EBV co-infected humanized mice

To investigate adaptive immune responses to KSHV in vivo, we infected human immune cell reconstituted NOD-scid IL2Rγc[−/−] (huNSG) or HLA-A2 transgenic huNSG (huNSG-A2) mice with KSHV and/or EBV and followed the infection for four weeks. (Fig. 1A). In addition to EBV wild-type (EBVwt, B95-8), we used a strain lacking the main lytic transactivator BZLF-1 (EBVzko) to limit EBV-specific T cell expansion, as T cell responses are frequently directed against lytic EBV antigens[24]. Moreover, lytic EBV replication is not essential for KSHV persistence in huNSG mice and may even be induced by KSHV[29,31]. Lymphoma formation was significantly increased in KSHV/EBV co-infected animals compared to all other infection groups, and B cells in dual-infected animals were found to harbor both viruses (Fig. S1)[29]. Splenic tissue sections of infected mice contained human CD3[+] T cells in close proximity to CD20[+] KSHV latency-associated nuclear antigen positive (LANA[+]) B cells, suggesting interaction between T cells and KSHV-infected B cells (Fig. 1B). Consistent with T cells responding to viral infection, numbers of peripheral blood and splenic CD3[+]CD8[+] T cells increased in all infected groups (Fig. 1C, D, S2, S2A-B), whereas B cells tended to decrease over the course of the experiment, even in control mice (Fig. 1C, S3A, B). The latter observation may be explained by the kinetics of human immune cell engraftment in NSG mice, which is skewed towards early B cell and late T cell development[32–34]. Interestingly, splenic and peripheral blood CD8[+] T cell numbers of KSHV/EBV co-infected mice at week 4 post-infection (p.i.) were higher compared to those solely infected with EBV, regardless of the EBV strain used (Fig. 1E, F, S3C, D). This may indicate an expansion of

not only EBV-specific, but also KSHV-specific T cells, upon co-infection with both viruses in this model. T cells did not expand in mice solely infected with KSHV, most likely due to the impaired persistence of KSHV in absence of EBV co-infection in humanized mice (Fig. S4A-D). The dendritic cell (DC) compartment only reconstituted in huNSG mice to a limited extent[35]. cDCs in blood increased upon KSHV/EBV co-infection, all other DC subsets did not expand upon infection (Fig. S5A, B). CD86, a co-stimulatory ligand on DCs that activates T cells upon binding, was not expressed at an increased level in KSHV/EBV co-infected animals (Fig. S5C). Additionally, maturation marker HLA-DR expression was also unchanged, revealing that maturation is not induced by KSHV/EBV infection (Fig. S5D).

### Limited KSHV-specific humoral responses are generated in humanized mice upon KSHV infection

Antibody responses in huNSG mice are generally weak, due to poor germinal center formation and the predominantly immature phenotype of the reconstituted B cell compartment[36]. However, we did document an increase in total IgM levels in the serum of infected mice (Fig. 2A). In line with inefficient class-switching in huNSG mice, serum IgG levels were greatly below the levels measured in human serum[37] and not significantly increased upon infection (Fig. 2B). In order to assess the presence of KSHV-specific antibodies, we tested huNSG mouse serum for IgM antibodies specific for 35, and IgG antibodies specific for 82 recombinant KSHV proteins, as described by[27]. While no specific IgG response was observed, we detected KSHV-specific IgM antibodies against a subset of proteins tested (Fig. 2C, D). The pattern of reactivity between KSHV/EBV co-infected mice was very heterogeneous, akin to the highly variable sero-reactivity in healthy and diseased KSHV carriers[27]. Among the most frequently recognized KSHV antigens were K5, K8.1, K10.5, K11, ORF6, ORF22, ORF25, ORF44, ORF50, and ORF73. Interestingly, among the ten IgM specificities that discriminated KSHV-positive from -negative mice were ORF73 (LANA) and K8.1, the two immunogenic proteins that are widely used during routine serological tests in humans[38] and that elicit responses in the majority of individuals with KSHV-associated malignancies[27].

Together, these data indicate that the humanized NSG model supports limited humoral responses in a subset of animals, which in part model the sero-reactivity of KSHV[+] individuals. The respective IgM responses also document broad viral protein expression upon KSHV infection of B cells in humanized mice.

### KSHV/EBV co-infection of humanized mice activates T cells and drives effector and central memory T cell differentiation

Viral infections can drive T cell activation and subsequent differentiation into effector and memory populations that play important roles in protection. Therefore, we examined expression of HLA-DR on human T cells as an indicator of activation. Both HLA-DR[+]CD4[+] and HLA-DR[+]CD8[+] T cell compartments expanded in the peripheral blood of EBV and KSHV/EBV, but not mock-infected or KSHV-only infected mice (Fig. 3A, S2, S3E, S4E, F).

Interestingly, CD8[+] T cell counts at the end of the experiment correlated with the number of activated (HLA-DR[+]) CD4[+] T cells in blood and spleen (Fig. S6A–D), suggesting that activated CD4[+] T cells help the observed expansion of CD8[+] T cell subsets.

Prior to infection, naïve CD45RA[+]CD62L[+] CD4[+] T cells comprised the largest proportion of human T cells in huNSG mice (Fig. 3B). In line with a more pronounced expansion of CD8[+] T cells, the ratio shifted towards CD8[+] T cells upon virus challenge (Fig. 3B). Furthermore, the frequency of cells with a naïve phenotype decreased upon infection, whereas the frequency with a memory phenotype increased, particularly in the CD8[+] T cell compartment (Fig. 3C). Similarly, infected mice, in particular KSVH/EBV dual-infected mice, had higher numbers of

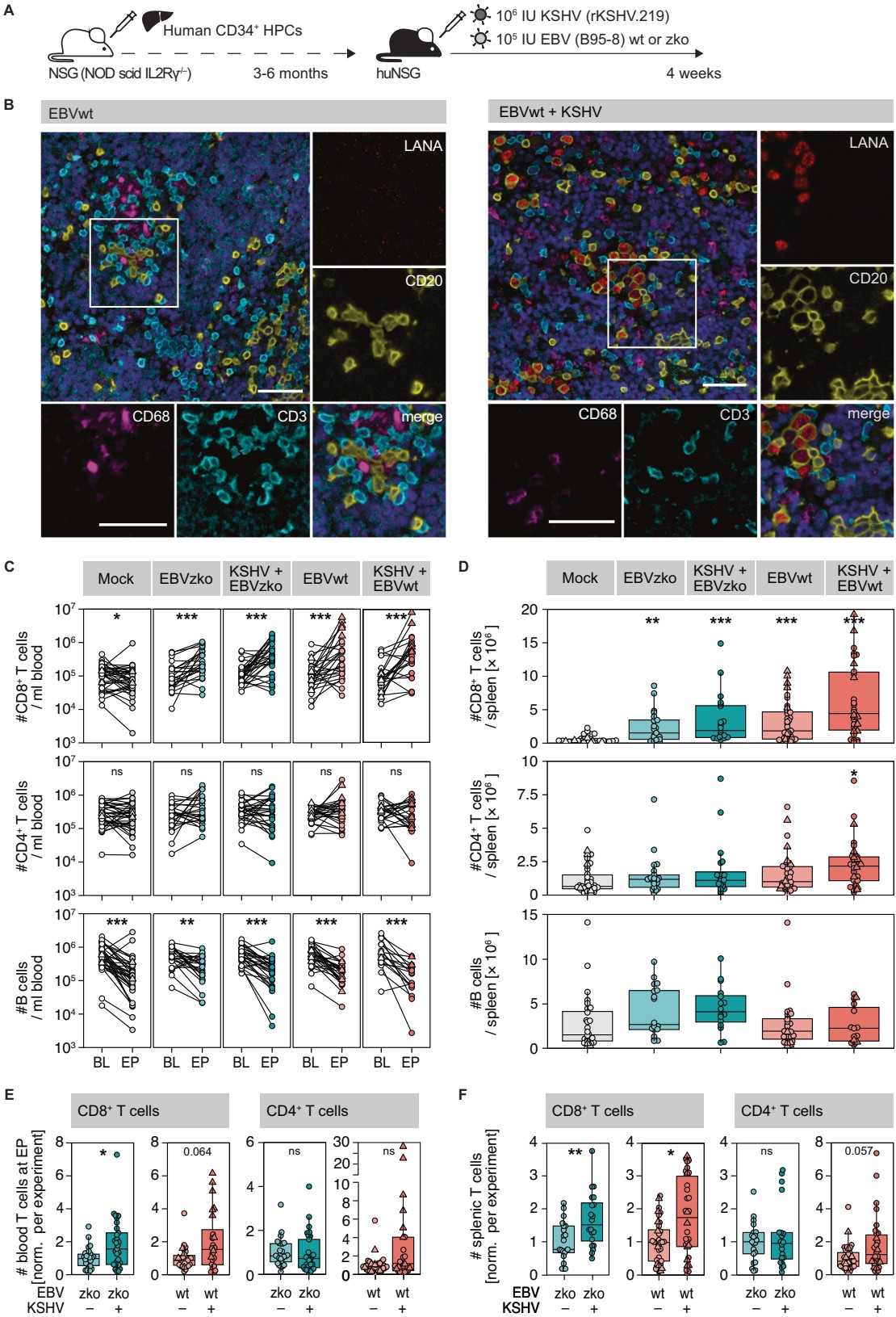

memory cells in the spleen when compared to mock infected animals (Fig. 3C).

Taken together, KSHV/EBV co-infection results in marked T cell activation and effector differentiation, which indicates the priming and expansion of EBV- and KSHV-reactive T cells upon infection, especially that of CD8⁺ T cells.

## T cells control viral titers and tumor load in KSHV/EBV co-infected humanized mice

In order to assess the importance of T cells in controlling KSHV in vivo, we depleted T cells prior to KSHV/EBV co-infection using antibodies directed against CD4 (OKT4) and CD8 (OKT8) epitopes, and followed the infection for 4 weeks (Fig. 4A, S8A). Previous studies have shown

**Fig. 1 | KSHV infection elicits an expansion of CD8+ T cells in EBV co-infected humanized mice. A** Experimental outline of EBV and KSHV infections in NSG mice reconstituted with human fetal liver derived CD34+ HPCs. Animals were infected intraperitoneally (i.p.) with KSHV and/or wild-type EBV (EBVwt), lytic replication deficient BZLF-1 knock-out EBV (EBVzko) or mock-infected with PBS at the age of three to six months and euthanized at 4 weeks post infection (p.i.) or when predetermined euthanasia criteria were met. **B** Co-staining of LANA with CD20, CD3 and CD68 on splenic sections of EBVwt (left) and EBVwt+KSHV (right) infected mice. **C** Human CD45+CD3+CD8+, CD45+CD3+CD4+ and CD45+CD19+ cells/ml blood measured before infection (baseline, BL) and at experiment termination (end point, EP). Composite data from 13 independent experiments for T cell data with $N = 35$ (mock), $N = 24$ (EBVzko), $N = 28$ (KSHV+EBVzko), $N = 26$ (EBVwt) and $N = 22$ (KSHV+EBVzko) mice; and from 12 independent experiments for B cell data with $N = 36$ (mock), $N = 24$ (EBVzko), $N = 29$ (KSHV+EBVzko), $N = 28$ (EBVwt) and $N = 17$ (KSHV+EBVwt) mice. Two sided Wilcoxon test. Significant p-values left to right: CD8: 0.04383; 5.329e-5; 5.53e-5; 0.0003636; 0.0006938; B cells: 6.657e-6; 0.002815; 1.193e-5; 5.245e-6; 3.052e-5. **D** Number of human CD45+CD3+CD8+, CD45+CD3+CD4+ and CD45+CD19+ cells per spleen. Composite data from 14 independent

experiments for T cells with $N = 34$ (CD8) and 38 (CD4) (mock), $N = 32$ (CD8) and 21 (CD4) (EBVzko), $N = 19$ (KSHV+EBVzko), $N = 36$ (EBVwt), $N = 30$ (KSHV+EBVwt) mice, and 10 independent experiments for B cells with $N = 28$ (mock), $N = 18$ (EBVzko), $N = 16$ (KSHV+EBVzko), $N = 27$ (EBVwt) and $N = 15$ (KSHV+EBVwt) mice. Kruskal Wallis followed by Dunn's test with Bonferroni (BF)-corrected p-values. Significant p values from left to right: CD8: 4.761e-3; 3.5799e-5; 6.577e-6; 2.116e-10; CD4: 0.006. **E** CD45+CD3+CD8+ and CD45+CD3+CD4+ cell numbers measured at 4 weeks p.i. in the blood or (**F**) spleen, normalized to the respective EBVzko or EBVwt single-infected group per experiment. Composite data of (**E**) 7 (EBVzko) and 8 (EBVwt) independent experiments with $N = 26$ (Ezko), $N = 29$ (KSHV+Ezko), $N = 27$ (Ewt), $N = 24$ (KSHV+Ewt). Two sided Mann Whitney U Test (MWU), significant p-values from left to right: 0.01297 (**F**) 6 (EBVzko) and 11 (EBVwt) independent experiments with $N = 21$ (Ezko), $N = 20$ (KSHV+Ezko), $N = 36$ (Ewt), $N = 30$ (KSHV+Ewt). Two sided MWU, significant p values from left to right: 0.008417; 0.03508. *$p < 0.05$, **$p < 0.01$, ***$p < 0.001$. **D**–**F** Box plot hinges correspond to 25th and 75th percentiles, shown are median and Turkey Whiskers. Source data are provided as a Source data file.

that EBV loads increase upon depletion of T cells in humanized mice over the course of EBV-single infection[20,22]. Similarly, we found that in absence of CD4+ and CD8+ cells blood titers of both EBV and KSHV significantly increased over the course of the experiment (Fig. 4B, C, S7B). T cell depletion on top of the co-infection did not coincide with a higher frequency of mice developing tumors despite higher viral titers, however, the number of tumors found in tumor-bearing mice was increased (Fig. 4D, S7C).

To further characterize the T cell responses elicited in this model, we performed TCR sequencing of T cells isolated from KSHV/EBV dual-infected huNSG-A2 mice. We found a broad spectrum of TCR sequences that differed greatly between individual mice (Fig. S8, S9). Mice did not share TCR sequences, which is in line with the lack of immunodominance observed in B and T cell responses among KSHV+ individuals in recent studies[25,27].

## KSHV-specific T cells are primed in KSHV/EBV co-infected humanized mice

Next, we aimed to analyse T cell specificities in CD19-negative splenocytes from KSHV/EBV co-infected animals using an IFNγ ELISpot with pools of peptides (overlapping 15-mers) spanning 82 KSHV proteins[25]. Initial screens revealed responses against KSHV peptide pools for ORF66, ORF44 and ORF73 (Fig. S10), however, we failed to expand these clones with the corresponding peptides.

Using an alternative approach, we then set out to isolate KSHV-specific T cells by capturing cells specifically activated upon contact with KSHV-infected B cells. Indeed, CD19-negative splenocytes from some KSHV/EBVzko co-infected animals preferentially produced IFNγ in response to autologous KSHV/EBVzko co-infected B cells (KE-CL) compared to EBVzko single-infected counterparts (E-CL), indicating that KSHV/EBV co-infected humanized mice support the priming of KSHV-specific T cells (Fig. 5A, S11A, B).

Expansion of cells from animals with KE-CL reactivity and subsequent co-culture with autologous KE-CLs for 24 h enabled us to enrich for reactive T cells by capturing IFNγ producing cells (Fig. 5B). Limiting dilution of these reactive T cells generated four T cell subpopulations, two CD4+ (10-01, 10-14) and two CD8+ (01-1, 1-9), that produced IFNγ almost exclusively upon co-culture with KE-CLs suggesting reactivity against KSHV proteins presented by KE-CL but not E-CL (Fig. 5C). Analysis of a repertoire of 22 TCR Vβ sequences by flow cytometry revealed that the isolated T cell subpopulations have a high degree of clonality, indicating that each clone derived from a single KSHV-reactive cell (Fig. S12, Table S1). Similar to IFNγ production, CD4+ and CD8+ T cell subpopulations degranulated more readily upon contact with autologous KE-CL but not E-CL, as measured by CD107a surface levels (Fig. 5D, S13A). However, only CD8+ T cell

subpopulations preferentially killed KE-CL, with 01-1 being the most effective (Fig. 5E, F, S13B). To evaluate killing capacity in vivo, we injected T cells (10-1, 10-14, 01-1) together with autologous E-CLs labeled with cell trace far red and KE-CLs labeled with cell trace blue into non-reconstituted NSG mice and assessed the ratio and killing of these cells after 5 h (Fig. S15A). In line with the in vitro results, we observed preferential killing of KE-CL cells by the CD8+ T cell subpopulation 01-1 in vivo, but not by the CD4+ T cell subpopulations tested in this assay (Fig. S15B). By probing against KSHV proteome wide peptide pools of 15mer-overlapping peptides[25], we found that only the K6 peptide pool was able to elicit a response from the T cells isolated by the IFNγ capture assay (Fig. 5G, Table S2). In line with this, CD8+ T cell subpopulation 01-1 produced significantly higher IFNγ levels upon co-culture with K6 peptide pulsed compared to unpulsed KE-CL (Fig. 5H), indeed suggesting specificity to K6.

Altogether, our data demonstrate that KSHV/EBV co-infected huNSG mice can support the generation of KSHV-specific T cell and humoral responses. We found that depletion of T cells led to an increase in KSHV titers and tumor burden. Furthermore, we identified K6-specific CD8+ T cells as effective in killing KSHV-infected B cells, suggesting K6 as a KSHV antigen that warrants further studies on its function in protection against KSHV-associated lymphomas.

## Discussion

In this study, we investigated human adaptive immune responses against KSHV infection of B cells in a humanized mouse model. These mice can robustly be infected with KSHV, provided they are EBV co-infected, and harbor KSHV in EBV-infected B cells[29]. Here, we report strong evidence for KSHV-specific human T and B cell responses in EBV co-infected humanized NSG mice. We identified antibody responses against ORF73 and K8.1, amongst others, and could identify ORF66, ORF73 and K6 as T cell targets. K6 could be further identified as a target for cytotoxic CD8+ T cells that were able to eliminate KSHV infected B cells, suggesting K6 as a T cell antigen that could be incorporated into KSHV-specific vaccination approaches. This is supported by a recent patient study that, while lacking evidence for immunodominant viral antigens, reported one individual with a dominant K6 specific T cell response and 20-30% of patients with a detectable immune responses against ORF73 and K8.1[39].

During primary EBV infection, CD8+ T cells expand to high numbers in adolescents with IM[40] and in humanized mice[20,23], exceeding the magnitude of CD4+ T cell expansion. The majority of EBV-specific T cells in the early infection phase in humans have been described to react to lytic antigens[41]. The expansion of CD8+ T cells upon EBV/KSHV infection surpassed the expansion observed in EBV-single infected mice, implying proliferation of KSHV-specific cytotoxic T cells. The

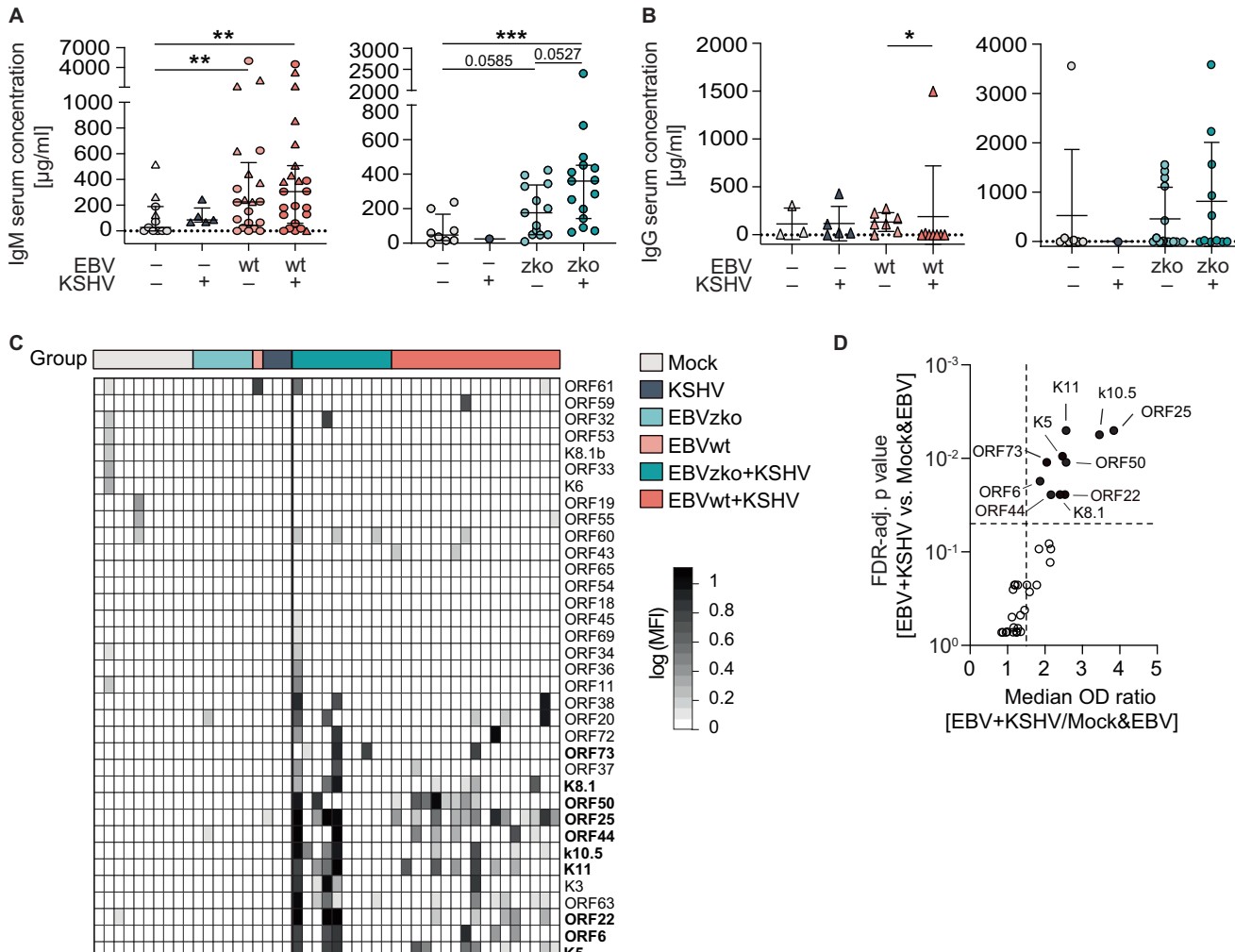

**Fig. 2 | Limited KSHV-specific humoral responses are generated in humanized mice upon KSHV infection. A** IgM serum levels of humanized NSG mice 4 weeks after i.p. infection with mock, KSHV only, EBV (wt or zko) only or EBV (wt or zko) and KSHV. 11 (EBVwt) independent experiments with serum from $N = 11$ (mock), $N = 5$ (KSHV), $N = 21$ (EBVwt), and $N = 23$ (KSHV+EBVwt) mice and 6 (EBVzko) independent experiments with $N = 8$ (mock), $N = 1$ (KSHV), $N = 13$ (EBVzko) and $N = 15$ (KSHV+EBVzko) mice. Data are presented as individual data points with mean and SD. MWU **$p < 0.01$, ***$p < 0.001$. Significant p values: mock – Ewt 0.0099; mock – Ewt+K 0.0023; mock – Ezko+K 0.0008. **B** IgG serum levels of humanized NSG mice 4 weeks after i.p. infection with mock, KSHV only, EBV (wt or zko) only or EBV (wt or zko) and KSHV. 3 (EBVwt) independent experiments with serum from $N = 3$ (mock), $N = 5$ (KSHV), $N = 7$ (EBVwt), and N = 8 (KSHV+EBVwt) mice 6 (EBVzko) independent experiments with $N = 7$ (mock), N = 1 (KSHV), N = 14

(EBVzko) and $N = 11$ (KSHV+EBVzko) mice. Data are presented as individual data points with mean and SD. Two sided MWU *$p < 0.05$. Significant p value Ewt – Ewt+K 0.0126 (**C**) Heatmap of KSHV-specific IgM antibodies in serum from humanized mice 4 weeks after i.p. infection with mock, KSHV, EBV (wt or zko) or EBV (wt or zko) and KSHV, measured by ELISA. Columns represent mice, rows represent antigens. Color intensity represents the background-substracted optical density (OD) on a logarithmic scale. **D** Ratio of median sero-reactivity (OD) of EBV + KSHV and Mock&EBV per antigen is indicated on the x-axis with the corresponding FDR-adjusted p-value on the y-axis. Labels indicate antigens with a significant ($p < 0.05$) and greater than 1.5 fold sero-reactivity in EBV⁺KSHV⁺ vs Mock&EBV⁺ mice. **C**, **D** Composite data with 47 mice from 4 independent experiments, $N = 10$ (mock), $N = 3$ (KSHV), $N = 6$ (Ezko), $N = 10$ (Ezko+K), $N = 1$ (Ew), $N = 17$ (Ew+K) mice per group. Source data are provided as a Source data file.

CD4⁺ T cell compartment, on the other hand, only reacted with activation and memory differentiation to the presence of KSHV. Although cytotoxic T cells are considered to be the major player in keeping EBV-infected cells under control, EBV-specific CD4⁺ T cells expand during primary infection and exert helper functions or even direct killing of infected B cells[42], and control viral titers in EBV-infected humanized mice[20]. Our model supported the priming of both KSHV-specific CD4⁺ and cytotoxic CD8⁺ T cells, supporting observations in healthy donors and patients with malignancies where both CD4⁺ and CD8⁺ KSHV-specific responses are observed.

CD4⁺ T cell expansion in response to KSHV was poor, which might be due to the downregulation of MHC class II molecules on the surface of EBV/KSHV-infected B cells, impeding the direct recognition of infected cells by CD4⁺ T cells.

Herpesviruses have evolved evasion strategies that interfere with antigen presentation via MHC class II molecules. The human cytomegalovirus (HCMV) protein US2, for example, promotes MHC class II endocytosis and proteasomal degradation[43], HCMV US3 interferes with MHC class II assembly and peptide loading[44], and EBV gp42 sterically blocks the interaction of the MHC class II/peptide complex with the TCR[45].

Most strikingly, KSHV and EBV both target MHC class II expression directly at the level of transcription by interfering with the expression and activity of its master transcription factor, the class II transactivator (CIITA). Mechanisms include the direct inhibition of CIITA promoter activity by EBV BZLF-1 binding to the promoter region[46]. EBV LMP2A and KSHV LANA indirectly inhibit CIITA by interfering with positive regulators E47 and PU.1 or IRF-4, respectively. KSHV vIRF3 can lead to a

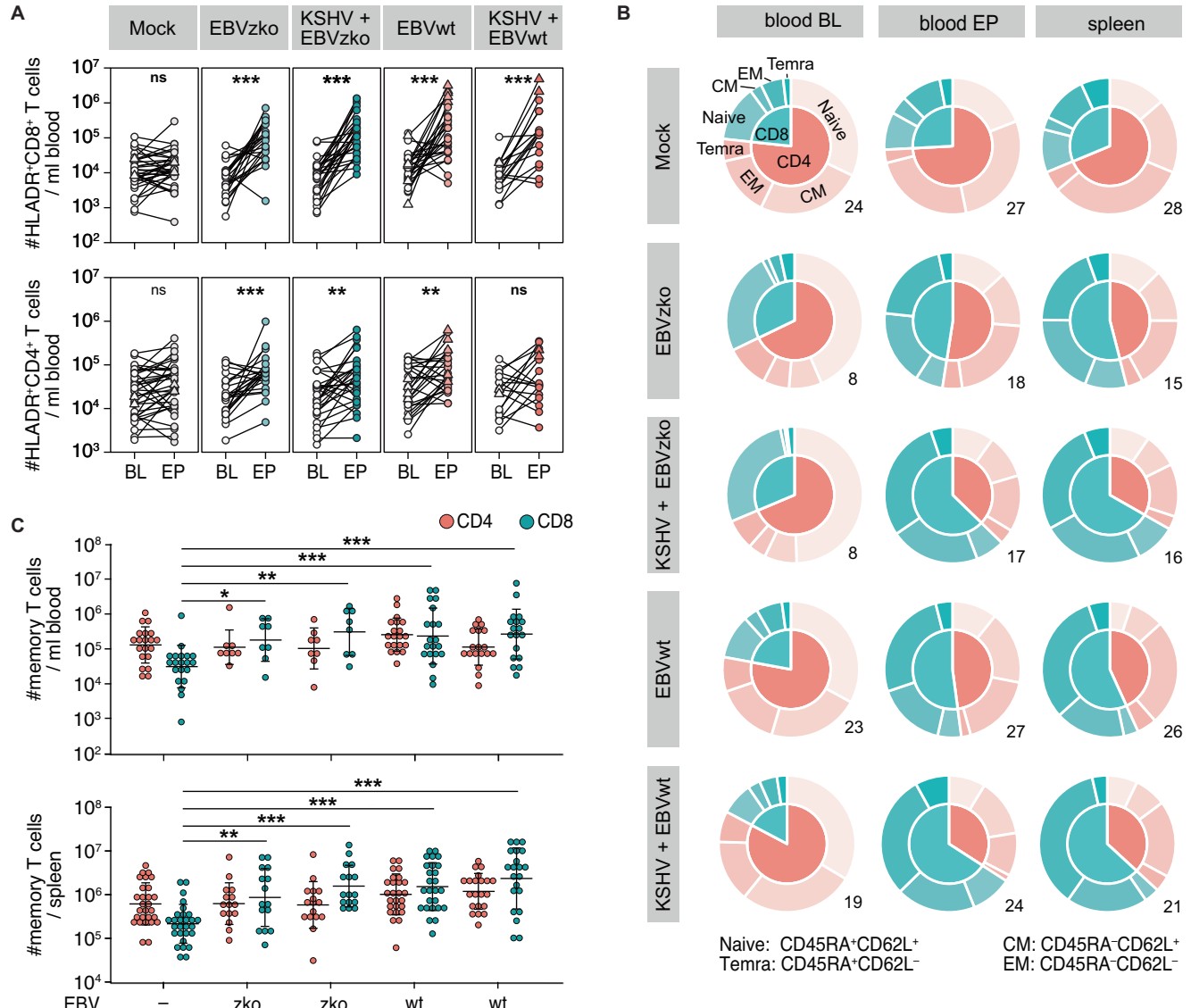

**Fig. 3 | KSHV/EBV co-infection of humanized mice activates T cells and drives effector and central memory T cell differentiation. A** Human CD45⁺CD3⁺CD8⁺HLA-DR⁺ and CD45⁺CD3⁺CD4⁺HLA-DR⁺ cells / ml blood at baseline (BL) and at experimental end point (EP). Composite data from 12 independent experiments with $N = 35$ (mock), $N = 24$ (EBVzko), $N = 28$ (KSHV+EBVzko), $N = 26$ (EBVwt) and $N = 17$ (KSHV+EBVwt) mice. Two sided Wilcoxon test. Exact $p$-values left to right: CD8⁺HLA-DR: 0.1196; 1.744e-6; 6.149e-9; 1.895e-5; 2.889e-5; CD4⁺HLA-DR: 0.05115; 0.0001503; 0.001673; 0.00148; 0.06045. **B, C** Composite data of 8 (blood) and 10 (spleen) independent experiments with $N = 20$ (blood) and 28 (spleen) (mock), $N = 8$ (blood) and 15 (spleen) (EBVzko), $N = 8$ (blood) and 16

(spleen) (KSHV+EBVzko), $N = 20$ (blood) and 26 (spleen) (EBVwt) and $N = 18$ (blood) and 21 (spleen) (KSHV+EBVwt) mice. Naive: CD62L⁺CD45RA⁺, central memory (CM): CD62L⁺CD45RA⁻, effector memory (EM): CD62L⁻CD45RA⁻, effector memory RA⁺ (Temra): CD62L⁻CD45RA⁺. **B** Differentiation status of T cells in peripheral blood and spleen presented as pie charts. **C** Number of T cells with Temra, EM or CM phenotype in peripheral blood and spleen at end point. Data are presented as individual data points with mean and SD. Two sided unpaired t-test. Significant $p$-values left to right: Blood: 0.040467; 0.00419; 0.000665; 0.000413; Spleen: 0.00475; 1.64e-5; 1.01e-6; 2.34e-8. *$p < 0.05$, **$p < 0.01$, ***$p < 0.001$. Source data are provided as a Source data file.

reduction of IFNγ levels, thereby affecting IFNγ-inducible CIITA-PIII promoter activity[47], which also affects MHC class II presentation through CIITA-independent mechanisms[48]. Furthermore, CIITA expression is repressed by the B lymphocyte–induced maturation protein 1 (BLIMP-1) in plasma cells[49], and most BLIMP-1⁺ lymphoma with a plasma cell-like phenotype, such as plasmablastic lymphoma (PBL) and PEL, completely lack MHC class II molecules or present with low surface levels[47,50]. Accordingly, PEL cell lines are only poorly recognized by LANA-specific CD4⁺ T cells[51]. Further, KSHV can directly inhibit MHC class II presentation through RTA-driven MARCH8 expression that increases MHC class II proteasomal degradation and

viral Bcl2 that blocks autophagy required to process antigens for MHC class II display[4,52–54].

In contrast to CD4⁺ T cells, CD8⁺ T cells expanded massively upon KSHV infection. Using a proteome-wide screening approach to test for KSHV specificities, we found evidence for reactivity against K6, a lytic gene that encodes for the viral macrophage inflammatory protein I (vMIP-I), a chemokine that acts as a specific agonist for the CC chemokine receptor (CCR) 8[55,56]. Similarly, KSHV-specific T cells in humans predominantly target lytic proteins[25,26], yet no immunodominant epitopes have been described so far. We were able to show that K6-specific CD8⁺ T cells have the potential to specifically kill

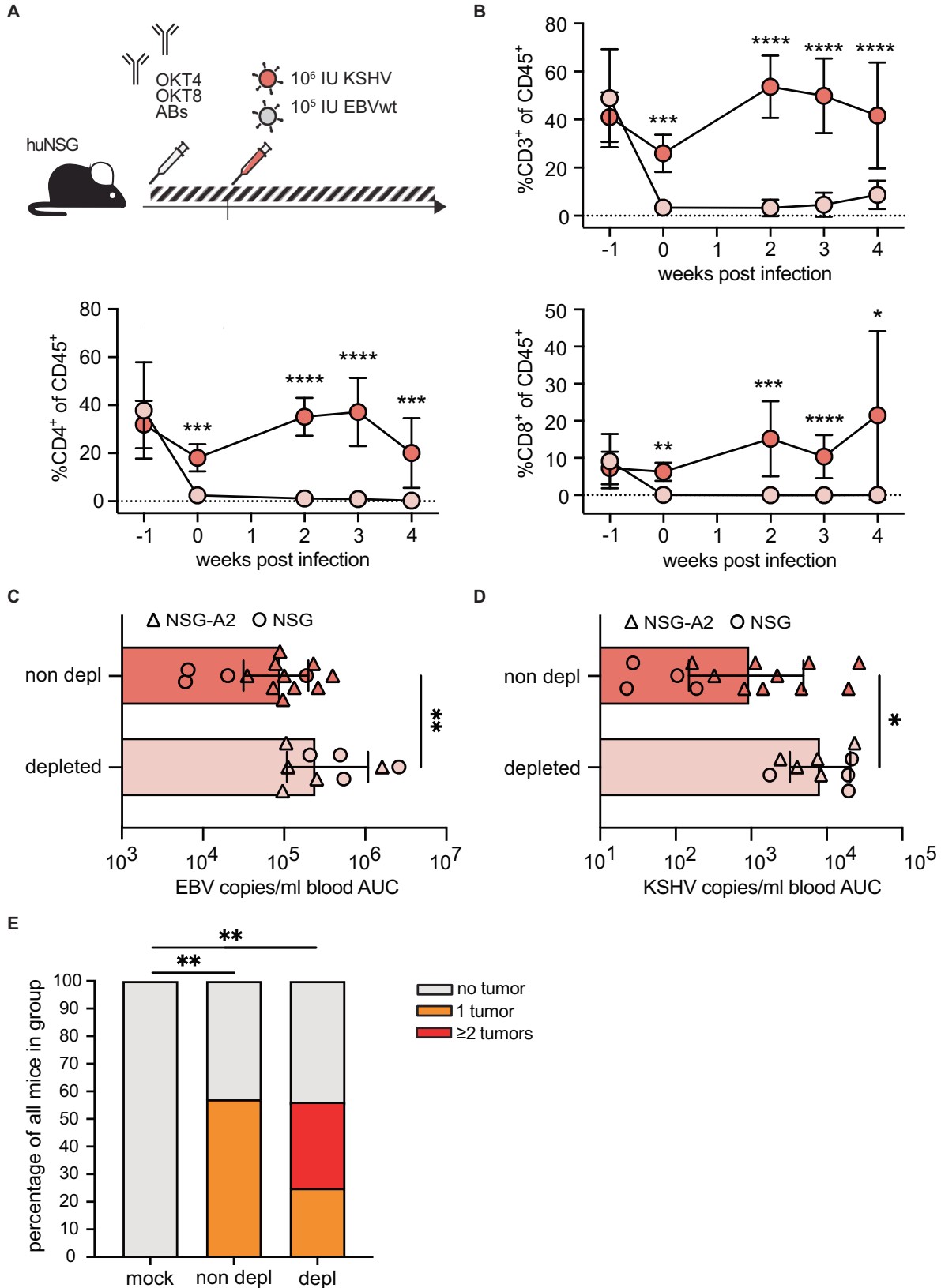

KSHV-infected B cells both in vitro and in vivo. Future efforts will focus on the exact epitope-deconvolution and HLA restriction of KSHV-specific T cells, as well as their protective value during virus challenge in vivo. Along these lines it is remarkable that gene copy number amplifications have been found in the K5-K6 region of KSHV genomes sequenced from Kaposi sarcoma[55]. These viral genome variations were not detected in KSHV shed into the saliva by the respective patients. Increased K5 and K6 transcript expression was associated with less disseminated disease[55], and could therefore allow for residual immune control even in HIV co-infected patients. Intratumorally elevated expression might also suggest K6 as a promising T cell antigen to target the respective Kaposi sarcomas. Such KSHV antigens that elicit

**Fig. 4 | T cell depletion increases viral loads and tumor burden of KSHV/EBV co-infected mice. A** Experimental outline of KSHV EBV co-infections with or without T cell depletion by i.p. injection of αCD8 (OKT8) and αCD4 (OKT4) antibodies. **B** Percentage of CD45+ CD3+, CD45+ CD4+, CD45+ CD8+ T cells in blood throughout the course of the experiment comparing mice with and without T cell depletion. Data present mean and SD. Two way mixed effects model with Geisser-Greenhouse correction, for comparisons Bonferroni's multiple comparison test was performed. Significant *p*-values left to right: CD3: 0.0006; < 0.0001; < 0.0001; < 0.0001. CD4: 0.0007; < 0.000; < 0.0001; 0.0002; CD8: 0.0024; 0.0002; < 0.0001; 0.0129. **C** EBV

copies/ml blood and (**D**) KSHV copies/ml blood displayed as area under the curve (AUC) during the experiment for KSHV/EBV co-infected mice with and without T cell depletion. Mean with SD, two sided MWU; Exact *p*-values are (**C**) 0.0086 and (**D**) 0.0252 (**E**) Tumor load of mock and KSHV/EBV co-infected animals with and without T cell depletion. Two sided MWU. Exact p values mock – non depl 0.0022; mock – depleted 0.0028. Composite data of 3 independent experiments (2 NSG-A2, 1 NSG) with *N* = 25 (non-depleted) and *N* = 13 (depleted) mice in (**B**) and *N* = 13 (non-depleted) and *N* = 9 (depleted) mice in (**C, D**). *\*p* < 0.05; \*\**p* < 0.01; \*\*\**p* < 0.001; \*\*\*\**p* < 0.0001. Source data are provided as a Source data file.

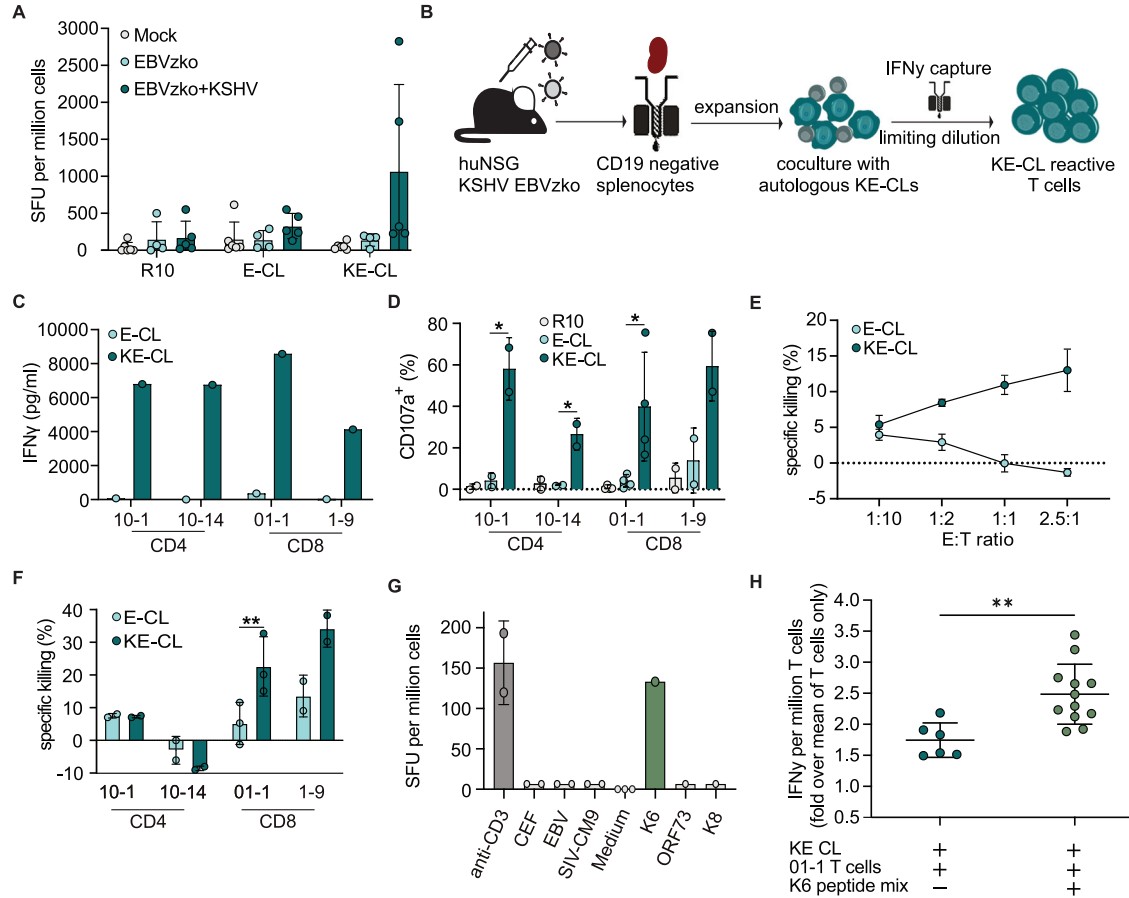

**Fig. 5 | KSHV-specific T cells are primed in KSHV-infected humanized mice.**
**A** IFNγ ELISpot of CD19-negative cells isolated from humanized mice after overnight culture with medium (R10), E-CL or KE-CL. Shown as spot forming units per one million CD19-negative cells; data are presented as individual values for R10 and mean values from duplicates for E-CL and KE-CL, with mean and SD. 1 experiment with *N* = 6 (Mock), *N* = 4 (EBVzko) and *N* = 5 (EBVzko+KSHV) animals. **B**. Experimental outline of the T cell isolation protocol from KSHV/EBV co-infected huNSG mice. CD19-negative splenocytes from KSHV±/EBVzko± mice were co-cultured with autologous KE-CLs for 24 h, IFNγ producing cells were isolated by IFNγ capture assay and cultured in limiting dilution. **C** IFNγ levels (pg/ml) in supernatant of isolated CD4+ and CD8+ T cell subpopulations after co-culture with E-CL or KE-CL were determined by ELISA upon growth after single cell dilution. *N* = 1.
**D** Degranulation measured by % CD107a positivity on the isolated T cell subpopulations after co-culture with R10, E-CL or KE-CL. *N* = 2 (10-1, 10-14 and 1-9), *N* = 4 (01-1) independent experiments, means of technical duplicates and SD are shown. Two sided unpaired *t*-test, *\*p* < 0.05; Significant p values 10-1 0.0393; 10-14 0.0235; 01-01 0.0369. **E** Specific killing (% dead cells compared to target cell without T cell condition) measured for CD8+ T cell subpopulation 01-1 after co-culture with E-CL

or KE-CL at different effector to target cell ratios. 1 independent experiment, Shown are means of 2 technical replicates with SD. **F** Specific killing measured for CD4+ and CD8+ T cell subpopulations after co-culture with E-CL or KE-CL at an effector-target ratio of 2.5 to 1. 2 (10-1, 10-14, 1-9) or 3 (01-01) independent experiments, individual values represent means of technical replicates, bars show means, error bars represent SD. Two sided MWU, \*\**p* < 0.01. exact *p*-value 01-01 0.0043 (**G**) Number of spot forming units per one million cells measured via IFNγ KSHV proteome wide ELISpot of KSHV-reactive T cells (isolated from the IFNγ capture assay) after stimulation with overlapping peptide pools for the indicated KSHV or control antigens. Figure shows the individual values from all wells with a signal (all data can be found in Supplementary Table 2) with mean and SD. **H** Relative IFNγ production of the T cell subpopulation 01-1 after co-culture with KE-CL or KE-CL pulsed with the K6 peptide pool. *N* = 2 independent experiments, *n* = 3 (KE-CL + T cells) and *n* = 6 (KE-CL + T cells + K6 peptide mix) replicates per experiment. IFNγ production relative to the mean of the T cell only condition is shown for each replicate, SD is displayed. Two sided Unpaired *t*-test, \*\**p* < 0.01. exact *p*-value 0.0033. Source data are provided as a Source data file.

such protective T cell responses could then be incorporated into KSHV-specific vaccine formulations to convert high seroprevalence regions with significant pathogenesis by this human tumor virus into low seroprevalence regions[57].

## Methods

### Key resources table

See supplementary table 3.

### Contact for reagent and resource sharing

Further information and requests for resources and reagents should be directed to and will be fulfilled by the lead contact, Prof Christian Münz, PhD (christian.muenz@uzh.ch).

### Materials and data availability

This study did not generate new unique reagents. Source data are provided with this paper in the source data file.

### Ethics statement and animal protocols

The use of human fetal liver tissue was approved by the cantonal ethical committee of Zurich, Switzerland (protocol no. KEK-ZH-Nr. 2010-0057 and KEK-StV-Nr.19/08). All described animal experimentation was reviewed and approved by the veterinary office of the canton of Zurich, Switzerland (116/2008, 148/2011, 209/2014, 159/2017, 212/2022, 213/2022).

### Humanized mouse generation and infection

NOD-scid $\gamma_c^{-/-}$ mice (NOD.Cg-Prkdcscid Il2rgtm1Wjl/SzJ or NSG; strain #005557) and HLA-A2 transgenic NSG (NOD.Cg-Mcph1Tg(HLA-A2.1) 1Enge Prkdcscid Il2rgtm1Wjl/SzJ or NSG-A2, strain #009617) mice were obtained from the Jackson Laboratories, bred and maintained at the Institute of Experimental Immunology, University of Zurich. Newborn mice were sublethally irradiated (1 Gy) and injected intrahepatically with $1–3 \times 10^5$ CD34+ hematopoietic progenitor cells (HPCs) from human fetal liver (HFL) tissue sourced from Advanced Bioscience Resources. Human CD34+ cells were isolated by positive selection for CD34 using magnetic cell separation from the fetal tissue according to the manufacturer's recommendations (Miltenyi Biotec)[20].

Three to six months after HPC injection, mice with successful human immune cell engraftment, as determined by flow cytometry of PBMCs for human CD45 (huCD45), huCD3, huCD19, huCD4, huCD8 and huNKp46, were intraperitoneally (i.p.) injected with $10^6$ Infectious Units (IU) of KSHV (rKSHV.219), $10^5$ Raji-infecting Units (RIU) of EBV (B95-8-GFP or B95.8-BZLF-1-KO-GFP), simultaneously injected with KSHV and EBV or mock-infected with PBS. For T cell depletion experiments, animals were i.p. injected with 150 µg αCD8 antibody (OKT8, BioXcell) and 300 µg αCD4 (OKT4, BioXcell) prior to infection until T cell depletion was achieved as verified by flow cytometry for huCD45, huCD3, huCD4, huCD8 and live dead marker. Starting at week 2 post infection, T cell depleted mice were i.p. injected with 100 µg αCD8 antibody (OKT8, BioXcell) and 150 µg αCD4 (OKT4, BioXcell) every two days until the end of the experiment.

Each experiment included mice reconstituted with HPCs from one single HFL donor and criteria for animal distribution into experimental groups included sex and human immune reconstitution parameters. Mice were euthanized by $CO_2$ inhalation followed by terminal heart puncture as secondary killing method four weeks post infection or when meeting predetermined euthanasia criteria that included severe weight loss and a scoring for general health and distress as determined in the animal experiment protocol approved by the cantonal veterinary office (protocol numbers 209/2014, 159/2019, 213/2020). Excluded from the study were virus-inoculated mice without detectable EBV BamHI W fragment or KSHV ORF26 DNA, respectively, in blood or spleen, and no positive signal for EBNA-2 or LANA, respectively, in FFPE tissue sections.

This study includes data from mice of 19 reconstituted NSG and NSG-A2 cohorts with 264 animals. Experiments comprise a total of 152 female mice and 112 male mice of $19.6 \pm 4.9$ weeks of age (Mean ± SD; range: 13–35 weeks). The median frequency (and IQR: 25th, 75th percentile) of huCD45+ cells of peripheral blood lymphocytes before infection was 79.1% (67.1%, 85.7%), huCD3+ T cells of huCD45+ lymphocytes: 34.6% (25%, 47.5%), huCD19+ B cells of huCD45+ lymphocytes: 57.1% (41.1%, 66.4%), huCD4+ cells of huCD3+ huCD45+ lymphocytes: 74.6% (68%, 80.1%), huCD8+ cells of huCD3+ huCD45+ lymphocytes: 22.8% (16.9%, 28.2%).

### Recombinant EBV and KSHV

EBV B95-8-GFP (EBVwt, p2089) and EBV B95-8-BZLF-1KO-GFP (EBVzko) were produced in HEK293 cells by inducing lytic gene expression through transfection with plasmids carrying BZLF-1 and BALF-4[58]. Recombinant KSHV (rKSHV.219) was produced in BrK.219 cells using anti-IgM antibody (0.625 µg/ml, Southern Biotech) and TPA (12-O-Tetradecanoylphorbol 13-acetate, 0.05 µg/ml, Sigma-Aldrich) adapted from[59,60], and from iSLKBac16.219 cells using doxycycline induction of an RTA plasmid carried by the iSLKs and sodium butyrate (1 mM, Sigma-Aldrich)[61]. EBV and KSHV concentrates were titrated on Raji or HEK293T cells, respectively, and GFP-positive cells were assessed 48 hours after in vitro infection on a FACSCanto II (BD Biosciences), as previously described in ref. 24.

### Quantification of viral DNA

DNA from whole blood was extracted using the NucliSENS EasyMag System (bioMérieux) and DNA from splenocytes was extracted using the DNeasy Blood & Tissue Kit (QIAGEN), respectively, according to the manufacturer's protocol. Viral DNA was quantified by Taqman (Applied Biosystems) real-time PCR as previously described in ref. 29 with the following primer and probe sequences: For EBV BamHI W fragment DNA (modified from ref. 62): 5′-CTTCTCAGTCCAGCGCG TTT-3′, 5′-CAGTGGTCCCCCTCCCTAGA-3′, 5′-(FAM)-CGTAAGCCAGAC AGCAGCCAATTGTCAG-(TAMRA)−3′; for KSHV ORF26 DNA (modified from ref. 63): 5′-GCTCGAATCCAACGGATTTG-3′, 5′-AATAGCGTGCCCC AGTTGC-3, 5′-(FAM)-TTCCCCATGGTCGTGCCTC-(BHQ-1)−3′. Samples were measured on an ABI Prism 7700 Sequence detector (Applied Biosystems) or on a C1000 Touch CFX384 Real-Time platform (Bio-Rad, Hercules, CA, USA) and analyzed in duplicates or triplicates with 5 µl extracted DNA starting with 2 min at 50 °C and 10 min at 95 °C, followed by amplification (95 °C for 15 s, 60 °C for 1 min, 50 cycles). Cq values were determined using the CFX-manager software (BioRad) with a regression algorithm.

### Cell isolation and generation of infected B cell lines

White blood cell counts in whole blood were determined with a hematocytometer (Beckman Coulter AcT Diff Analyzer) and subjected to erythrocyte lysis by ACK lysis buffer. Spleens were manually dissociated and filtered through a 70 µm cell strainer, followed by separation of mononuclear cells on Ficoll-Paque gradients by centrifugation at 1000 g for 25 min at room temperature (RT) (GE Healthcare). B cells were isolated by magnetic cell sorting using CD19 microbeads (Miltenyi Biotech) following the manufacturer's protocol. Peritoneal lavage was performed at sacrifice by injecting 5 ml PBS into the peritoneum and re-extracting it as previously described in ref. 64. Peritoneal cell suspensions or splenic CD19+ cells were cultured in RPMI 1640 medium supplemented with 10% FBS, 50U/ml penicillin-streptomycin and 1% glutamine (R10) in 96 well flat-bottom plates. Alternatively, CD19+ spleen cells were spinfected with KSHV at a multiplicity of infection (MOI) of 0.5−1 at 800 g and 4 °C for 30 min in R10. After that, EBV was added at a MOI of 0.1 − 0.25 and centrifuged for an additional hour before cells were transferred to a 37 °C humidified incubator with 5% $CO_2$. KSHV presence in established cell lines was assessed by qPCR for ORF26,

as described above, after isolation of DNA using DNeasy Blood & Tissue Kit (QIAGEN).

## Flow cytometric analysis

Cells were stained for surface markers with fluorescently-labeled, anti-human antibodies purchased from BioLegend unless otherwise stated: CD45 (Pacific Blue/HI30; BUV395/HI30, BD Bioscience), CD3 (PE/UCHT1 or BV785/OKT3), CD8 (PerCP/SK1 or BV650/SK1), CD4 (APC-Cy7/RPA-T4 or BV605/OKT4), CD19 (PE-Texas Red/SJ25-C1, Thermo-FisherScientific; PE-Cy7/HIB19 or AF700/HIB19), HLA-DR (FITC/L243; PE-Cy7/L243; PE CF594/G46-6), CD62L (PE-Cy7/ DREG-56; APC/DREG-56, BD Bioscience), CD45RA (BV510/HI100), CD14 (QDot 655/TuK4, Thermo Fisher Scientific), CD16 (BUV737/3G8, BD Biosciences), CD1c (PE-Cy7/L161), CD11b (BV711/ICRF44), CD11c (Alexa Fluor 700/B-ly6), CD86 (BV510/IT2.2), CD141 (BB700/1A4, BD Biosciences) and CD303 (BV421/V24-785/BD Biosciences). Cell viability was assessed using Fixable Viability dyes (Zombie Near-IR and Zombie Aqua, Biolegend). TCR Vbeta sequences were stained in several rounds using the following antibodies (all from Beckman Coulter): Vβ1/PE, Vβ2/PE, Vβ3/FITC, Vβ4/PE, Vβ5.1/FITC, Vβ5.2/FITC, Vβ5.3/PE, Vβ6.7/FITC, Vβ7.1/PE, Vβ8/FITC, Vβ9/PE, Vβ11/FITC, Vβ12/PE, Vβ13.1/PE, Vβ13.6/FITC, Vβ14/PE, Vβ16/FITC, Vβ17/FITC, Vβ18/PE, Vβ20/PE, Vβ21.3/FITC, Vβ22/FITC and Vβ23/PE.

Single cell suspensions were incubated with antibodies for 30 min at 4 °C, washed and fixed with 1% PFA. Acquisition and compensation were performed on BD FACS Canto II, BD LSR II Fortessa and BD FACSymphony A5 flow cytometers. Data was exported and analyzed using FlowJo software (version 10, TreeStar Inc). Graphs and pie charts were generated using Prism (GraphPad Software) or R software (version 4.1.1.).

## Isolation and maintenance of KSHV/EBV infected B cell line-specific T cells

CD19-negative splenocytes from KSHVr-219 B95-8-BZLF-1KO-GFP dual-infected humanized mice were isolated using magnetic beads for CD19 (Miltenyi) according to manufacturer's instructions. CD19− cells were expanded using the adapted rapid expansion protocol[65] through culture with irradiated human PBMCs (50 Gy, 200-fold from 3 donors), OKT-3 (30 ng/ml, Miltenyi) and recombinant IL-2 (Pepro-tech, 3000 U/ml) in 1:1 R10 and X-Vivo/10%FBS (Lonza) for up to 14 days. Expanded T cells were co-cultured for 24 h with KSHV/EBV co-infected, autologous B cells (KE-CLs) at a ratio of 5:1 and IFNγ-producing T cells were isolated using the IFNγ secretion assay cell enrichment and detection kit (Miltenyi) according to manufacturer's instructions. IFNγ positive T cells were cultured in limiting dilution with irradiated feeder cells (per 96-well plate well $1e^4$ autologous KE-CL, 50 Gy; $1e^5$ PBMCs, 20 Gy) in RPMI 1640 (Gibco) with 10% human AB serum (Corning) (H10). Recombinant IL-2 (Peprotech) was added from day 3 on at a concentration of 125 U/ml. Growing T cell sub-populations were continuously cultured in H10 with 125 U/ml IL-2 with addition of newly irradiated feeder cells (irradiated PBMCs and KE-CL) every 10-14 days.

## TCR beta-chain library preparation, immune repertoire sequencing and bioinformatics analysis

Total RNA was extracted from cryopreserved CD4+ or CD8+ T cells isolated from splenocytes of KSHV/EBVzko infected huNSG-A2 mice using a standard Trizol protocol. The first-strand cDNA synthesis was performed using the anchored switch 5′ rapid amplification of cDNA ends (5′RACE) PCR based commercial SMARTer® RACE 5′/3′ Kit (Takara Bio). Based on recommendations in the SMARTer Human TCR a/b profiling kit (Takara Bio) user manual, two rounds of seminested PCR were performed in succession to amplify cDNA sequences corresponding to the variable regions of TCR-β transcripts. The concentration of gel purified amplicons was measured on a Qubit

Fluorometric Quantification System (Invitrogen). Samples were diluted to a concentration of up to 12 nM. Considering the initial cell count of every single library, the amplicons of the second round PCR were pooled proportionally without changing the molarity. The pooled library was sequenced using the Miseq Reagent Kit v3 (600-cycle, Illumina). Raw sequence reads were processed using the MiXCR (version 3.0.13) software pipeline[66]. MiXCR pre-processed data were post-analyzed by using in house custom R and Python codes, VDJtools (version 1.2.1) and Immunarch to evaluate clonal expansions, clonality, clonotype overlap, clonotype tracking, and top clonal composition[67,68].

## T cell degranulation assay

T cell clones growing out after isolation by IFNγ capture assay following limiting dilution were co-cultured with R10 only, PMA/Iono or autologous E-CLs or KE-CLs labeled with PKH26-PE (Merck) according to manufacturer's instructions at a ratio of 1:1 in a round- or v-bottom plate. Antibody for CD107a-FITC (BD Biosciences, H4A3) was added and cells were incubated at 37 °C 5% $CO_2$ for 1 h. Brefeldin A (Invitrogen) was added at 1:2000 and cells were incubated another 5 h at 37 °C 5% $CO_2$. Cells were stained for surface markers CD3 (BV785/OKT-3, Biolegend), CD4 (BV605/OKT4, Biolegend), CD8 (BV650/SK1, Biolegend) and viability marker (Zombie Near-IR, Biolegend) for 30 min at 4 °C. Samples were fixed with 1% PFA and markers as well as degranulation (CD107a-positive) were assessed using BD Fortessa LSRII.

## T cell in vitro killing assay

T cell subpopulations isolated by IFNγ capture assay following limiting dilution were co-cultured with R10 only and autologous E-CLs or KE-CLs labeled with PKH26-PE (Merck) according to manufacturer's instructions for 15–17 h at different ratios. Additionally, PKH-labeled E-CLs or KE-CLs were cultured without T cells as control. Supernatant was collected and frozen at −20 °C for cytokine analysis and cells were stained with viability marker Zombie Near-IR (Biolegend) and assessed using the BD Fortessa LSRII. Specific killing was calculated by substracting % of dead cells in the E-CL or KE-CL only control.

## T cell in vivo killing assay

Autologous E-CLs and KE-CLs were labeled with Cell Trace Violet and Far Red and Cell Trace Violet and Blue (Invitrogen), respectively, according to manufacturer's instructions. 4e6 E-CLs, 4e6 KE-CLs and 2e6 T cells were injected into the peritoneum of non-reconstituted NSG mice. 5 h post transfer mice were euthanized, and peritoneal lavage was performed by injecting 5 ml PBS/2%FBS and extracting the cell solution from the peritoneum with a syringe. The cell suspension was stained for CD3 (BV785/OKT3, Biolegend) and cell viability (Zombie near-IR, Biolegend) and analyzed using the BD Fortessa LSRII.

## Enzyme-linked Immunosorbent Assays (ELISAs)

For serum, blood was collected by terminal cardiac puncture in BD Mictrotainer tubes and spun at 8000 g for 1–2 min. Serum was collected and frozen at −20 °C until use. Human IgM and IgG levels in the serum of humanized mice were quantified by Human IgM ELISA Kit (Abcam) and Human IgG ELISA Kit (Sigma) according to the manufacturer's instructions. KSHV-specific IgG was quantified using the HHV-8 IgG Antibody ELISA Kit (Advanced Biotechnologies Inc) according to the manufacturer's protocol. Additionally, serum was tested for IgG antibodies specific for 73 recombinant KSHV proteins, as previously described by[27]. Similarly, serum samples were screened for IgM reactivities against the following recombinant KSHV proteins: K8.1, ORF73, ORF38, ORF65, ORF25, ORF59, ORF6, ORF61, ORF19, k10.5, ORF60, ORF44, ORF63, K3, ORF54, ORF22, ORF55, ORF45, K5, ORF11, ORF69, ORF18, ORF50, ORF20, ORF33, ORF72, ORF36, ORF43, ORF37, K11, K8.1b, K6, ORF34, ORF32, ORF53. The background was set at the mean plus two times the SD of Mock controls.

Supernatant of T cells reactive to KSHV and EBV dual-infected B cells was collected after overnight co-culture (15–17 h) with EBV only or KSHV and EBV co-infected, autologous B cell lines or R10 as negative control and stored at −20 °C until use. Production of human IFNγ was assessed by the human IFNγ ELISA Flex (MabTech) according to manufacturer's protocol using a 1:1 or 1:3 dilution of the supernatants.

### IFNγ Enzyme-Linked Immunosorbent Assay (ELISA)

T cells were used 10-14 days after refeeding or restimulation with irradiated PBMCs and KE-CLs. 100'000 or 200'000 cells were co-cultured with equal volume of R10, PMA/Iono or equal amount of autologous E-CLs or KE-CLs or KE-CLs pulsed with K6-peptide pool (37 °C, 1 h, 1 mM peptide) over night and IFNγ ELISA was performed on the supernatant according to manufacturer protocol (MabTech). IFNγ secretion was calculated per million T cells and mean of the T cell only condition (R10) was calculated. KE-CL and KE-CL peptide pulsed conditions were normalized to the mean of the T cell only condition.

### IFNγ Enzyme-Linked ImmunoSpot (ELISpot) assays

CD19-depleted splenocytes from KSHV/EBVzko-infected mice were isolated at four weeks p.i. and frozen at −80 °C until use. Upon thawing, 100'000 cells were co-cultured with equal volume of R10, PMA/Iono or equal amount of autologous E-CLs or KE-CLs over night and IFNγ ELISpot was performed according to manufacturer protocol (MabTech).

### KSHV proteome wide Enzyme-Linked ImmunoSpot (ELISpot) assay screening

KE-CL reactive T cells isolated by IFNγ capture assay were frozen until use. Upon thawing, T cells were expanded with anti-CD3 and 10–14 days later, 150'000 cells were screened with an IFNγ ELISpot against a library of 15-mer peptides covering the KSHV proteome as described in ref. 25.

### Histological staining

Tissue sections were formalin-fixed and FFPE-embedded. Staining of EBNA-2 (clone PE2, Abcam), LANA (clone LN53, Clinisciences), CD20 (clone SP32, Cell Marque) and LMP1 (clone CS1-4, Abcam) was carried out on a Leica BOND-III automated immunohistochemistry system with diaminobenzidine (DAB) as substrate (Zytomed Systems, Berlin, Germany). Multiplex immunofluorescence (IF) staining was performed using Opal dyes and Spectral DAPI (FP1490) from PerkinElmer. Opal dyes 540, 620, 650 and 690 were used to detect CD20 (clone SP32, Cell Marque), LANA (clone LN53, CliniSciences), CD68 (clone 514H12, Leica Biosystems) and CD3 (Clone SP7, Diagnostic Biosystem RMAB005), respectively. Single stained samples were used to determine the level of spectral overlap. Staining was measured on a Vectra3 automated quantitative pathology imaging system and analyzed using Phenochart (v1.0) and InForm (v2.4.8) software (all from PerkinElmer).

### Statistical analysis

Normal distribution of data was assessed by the D'Agostino & Pearson omnibus test. Non-parametric data were analyzed by unpaired Mann-Whitney U test or paired Wilcoxon signed-rank test. Non-parametric data with more than two groups were compared by Dunn's multiple comparison test following Kruskall Wallis test. Parametric data was analyzed by unpaired t tests. P values were adjusted for multiple hypothesis testing using Bonferroni (BF) method. A *p*-value of <0.05 was considered statistically significant. Statistical analysis was performed with R (version 4.1.1) and GraphPad Prism 9 (GraphPad Software, San Diego, CA).

### Reporting summary

Further information on research design is available in the Nature Portfolio Reporting Summary linked to this article.

## Data availability

Data are available in the provided Source Data File. All TCR Sequencing data are available on the SRA data base, accession code PRJNA1109285. Source data are provided with this paper.

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

## Acknowledgements

This research was in part supported by Cancer Research Switzerland (KFS-4091-02-2017; KFS-4962-02-2020), KFSP-Precision[MS] and HMZ ImmunoTargET of the University of Zurich, the Cancer Research Center Zurich, the Vontobel Foundation, the Baugarten Foundation, the Sobek Foundation, the Swiss Vaccine Research Insitute, Roche, Novartis and the Swiss National Science Foundation (310030B_182827, 310030_204470/1, 310030L_197952/1 and CRSII5_180323). D.W. was funded by federal funds from the National Cancer Institute, National Institute of Health with contract numbers 75N91019D00024/ HHSN261201500003I. N.G. was supported by the Bundesministerium für Bildung und Forschung (BMBF 031A232), Genzyme, and the Forschungskommission of the Heinrich Heine University Düsseldorf, Germany. N.C. and L.R. were supported by career advancement grants from the University of Zurich (Forschungskredit, FK-18-026, FK-21-034). M.B. and D.M. were supported by MD-PhD fellowships from the Swiss National Science Foundation and the Swiss Academy of Medical Sciences (323630_199389; 323530_145247).

## Author contributions

Conceptualization, N.C., L.R., M.B., D.M., and C.M; Formal analysis, N.C., L.R., M.B., D.M., R.R., W.M., S.B., N.L.; Investigation, N.C., L.R., M.B., D.M., R.R., W.M., N.Z., S.B., M.T., D.B., M.B., J.R.; Resources, C.M., D.W. and N.G.; Writing – Original Draft, N.C and L.R.; Writing – Review & Editing, N.C., L.R., M.B., D.M. and C.M.; Visualization, N.C. and L.R.; Supervision, C.M., D.W. and N.G.; Funding Acquisition, N.C., L.R., M.B., D.M., N.G., D.W. and C.M.

## Competing interests

The authors declare no competing interests.
