## [Peer Review File · Nature Communications]

KSHV infection of B cells primes protective T cell responses in humanized miceREVIEWER COMMENTS

Reviewer #1 (Remarks to the Author):

Caduff N et al made co-infection of KSHV and EBV into humanized mice, leads Ig-M response and induction of memory CD4 and CD8 T cells, especially K6 targeting CD8 T cells. This study is important to show the effect of EBV co-infection, and the possibility of K6 as vaccine antigens.

The authors showed CD8 T cells were dominant after infection. Is there any possibility by using HLA-A2 transgenic mice ? Why the did not use human class I and class II transgenic mice?

Which population of B cells EBV and KSHV were infected? In a same population or different populations ? Any evidence of co-infection ?

Are the lymphoma developing in these mice ? If developing, which types of lymphoma is developing ? If not developing lymphoma by CD8 T cells, are lymphoma developing by anti-CD4 or anti-CD8 antibody treatment ?

If the authors treat the mice with anti-CD4 Ab, are cytotoxic CD8 T cells expanding ?

Could you find Dendritic cell population in humanized mice after infection ?

Reviewer #2 (Remarks to the Author):

This study provides an elegant characterization of immune responses, both cellular and humoral in mice with EBV and KSHV co-infection. It focuses on characterizing the difference between single and double infection, and teases out responses to KSHV. The mice are able to develop IgM to K8.1 and LANA, like in individuals with KS, and a new potential peptide important for CD8 T cell responses and cellular mediated killing of infected B cells, which is ORF K6, was identified. These findings are interesting and relevant, and provide potential leads for vaccine development. The following are specific comments/questions (minor):

- The paper by Nalwoga et al, Nat Comm 2021 is not cited or discussed (although the authors are among the co-authors of this manuscript, and only an older 2017 paper from the Whitby lab is cited). How do the authors explain the lack of cellular responses to ORF-K6 in people infected with KSHV in an endemic region? This article should be cited and this potential discrepancy discussed.

- Results, first paragraph, line 93 mentions infection with KSHV and/or EBV. The first figures only show EBV +/- KSHV, but KSHV alone is not shown until the antibody section (Figure 3). Was there a reason for not showing the T cell responses for KSHV+/EBV- mice? Or it wasn't done?

Reviewer #3 (Remarks to the Author):

Kaposi sarcoma-associated herpesvirus (KSHV) and Epstein-Barr virus (EBV) are oncogenic viruses that contribute to between 1-2% of all cancers worldwide. Despite this significant burden, adaptive immune responses against these pathogens, in particular KSHV, have not been extensively investigated. This study by Caduff et al. is important as it investigates antigen-specific control of KSHV infection using a humanized mouse model. The authors describe very nicely cytotoxic T cell effector and memory phenotype expansion as well as the viral antigens recognized. Importantly, the authors show that KSHV K6 reactive CD8 T cells are capable of killing KSHV-infected B cells. This finding is significant as it raises the possibility to leverage K6 therapeutically and drive protective T cell responses. Overall, this is an important manuscript and is well controlled. This will be a study of high importance to the oncogenic virus community.

I only have one minor criticism of the manuscript. The identification of K6 reactive CD8 T cells is interesting and important and should be further discussed. It would be useful to have an expansion of the discussion on K6 genetic diversity in the discussion. Perhaps this genetic diversity is associated with disease progression or treatment outcomes.

Point-by-point response

We thank all three reviewers for their constructive comments which we have now all addressed. We high-light the changes below and with red font in the revised manuscript text.

Reviewer #1

The authors showed CD8 T cells were dominant after infection. Is there any possibility by using HLA-A2 transgenic mice? Why the did not use human class I and class II transgenic mice?

Some of the experiments were performed in humanized HLA-A2 transgenic mice. We marked the HLA-A2 transgenic mice with a different symbol (Triangle) in the revised manuscript version (Figure 1, Figure 2, Figure 3, Figure 4).

We also provide a comparison for T cell activation and expansion of KSHV and EBV infected humanized mice with and without HLA-A2 transgene in the new supplementary figure 3. While we see that NSG-A2 animals show increased CD8⁺ and CD4⁺ T cell populations in blood, NSG animals show increased CD8⁺ and CD4⁺ T cell populations in spleen. As both NSG and NSG-A2 cohorts show increases in splenic CD8⁺ T cells, the difference in NSG and NSG-A2 animals might be due to confounding factors that increase variability in the mock infected animals compromising normalization per experiment.

Which population of B cells EBV and KSHV were infected? In a same population or different populations? Any evidence of co-infection?

This has previously been shown by McHugh et al, 2017 for ex vivo cultured cell lines derived from infected animals. We provide histological stainings of liver and spleen sections from KSHV/EBV infected animals. Staining for EBV or KSHV nuclear antigens with CD20 shows that both EBV and KSHV infect B cells. Further, EBV and KSHV protein co-stainings show that both EBV and KSHV can infect the same cell (new supplementary figure 1).

Are the lymphoma developing in these mice? If developing, which types of lymphoma is developing? If not developing lymphoma by CD8 T cells, are lymphoma developing by anti-CD4 or anti-CD8 antibody treatment?

We now report lymphoma formation in EBV single and KSHV co-infected humanized mice in the new supplementary figure 1. During the submission period we had also performed three experiments in which T cell depleted (CD4 plus CD8 depleted) mice were compared to untreated controls during KSHV and EBV co-infection, and we report viral loads and lymphoma formation upon T cell depletion in the new Figure 4 and the new supplementary figure 7.

If the authors treat the mice with anti-CD4 Ab, are cytotoxic CD8 T cells expanding?

Prior studies with the murine γ 2-herpesvirus MHV-68, more closely related to the human γ 2-herpesvirus KSHV than to the human γ 1-herpesvirus EBV, have shown that CD8⁺ T cell expansion in the bronchoalveolar lavage with lung at the primary site of MHV-68 infection occurs to a similar degree with and without CD4⁺ T cell help (Cardin et al., J Exp Med 1996). This expansion is primarily dependent on infection induced inflammation while long-term CD8⁺ T cell function strongly depends on CD4⁺ T cell help. Accordingly, viral loads were similar in lung and adrenal gland during primary infection but increased continuously due to failing immune control from day 30 after infection without CD4⁺ T cell help.

In order to address this reviewer's comment, we now report correlation of activated, CD4⁺ HLA-DR⁺ cell numbers with CD8⁺ T cell numbers in blood and spleen (new supplementary figure 6), suggesting that CD4⁺ T cell activation does indeed seem to sustain CD8⁺ T cell expansion in our mouse model, similar to the late time point during MHV-68 infection. We also now

demonstrate that T cell depletion in general increased viral loads of both EBV and KSHV, resulting in elevated numbers of tumors found in lymphoma bearing mice.

Could you find dendritic cell population in humanized mice after infection?

Following this reviewer's recommendation, we performed multiparametric flow cytometry analysis of dendritic cell populations in uninfected and KSHV/EBV double-infected humanized mice (new supplementary figure 5). While present, dendritic cell numbers in humanized mice only increase for cDC2s and monocyte-derived DCs in blood. They show no difference in CD86 or HLA-DR expression and therefore do not show increased maturation between uninfected and infected animals. Hence, we hypothesize that while DCs might play a role in T cell priming, B cells that are infected are the most likely contributors to the significant T cell expansions that we observed.

Reviewer #2

- The paper by Nalwoga et al, Nat Comm 2021 is not cited or discussed (although the authors are among the co-authors of this manuscript, and only an older 2017 paper from the Whitby lab is cited). How do the authors explain the lack of cellular responses to ORF-K6 in people infected with KSHV in an endemic region? This article should be cited and this potential discrepancy discussed.

We discuss and cite the Nalwoga et al., Nat Commun 2021 study in the revised manuscript version. Indeed, one individual with a dominant K6 specific T cell response was reported in this study. No immunodominance for particular viral antigens was reported in the analyzed patient cohort but 20-30 % of the tested individuals showed detectable immune responses against ORF73 and K8.1, for which we detected antibody responses in the KSHV and EBV co-infected mice. Similarly, we provide now additional supplemental data that in other KSHV and EBV co-infected mice ORF44, ORF66 and ORF73 specific responses were detected in splenic bulk T cell populations (new supplementary figure 10). However, we were only able to characterize K6 specific CD8⁺ T cells and their protective function at the clonal level.

- Results, first paragraph, line 93 mentions infection with KSHV and/or EBV. The first figures only show EBV +/- KSHV, but KSHV alone is not shown until the antibody section (Figure 3). Was there a reason for not showing the T cell responses for KSHV+/EBV- mice? Or it wasn't done?

We now included the new supplementary figure 4 providing the characteristics of T cell expansion and activation for KSHV single infection. Similarly to lack of reactivity at the antibody level in mice that had been injected with KSHV alone, also no T cell activation and expansion was observed. This is most likely due to lack of KSHV persistence without EBV co-infection.

Reviewer #3

Kaposi sarcoma-associated herpesvirus (KSHV) and Epstein-Barr virus (EBV) are oncogenic viruses that contribute to between 1-2% of all cancers worldwide. Despite this significant burden, adaptive immune responses against these pathogens, in particular KSHV, have not been extensively investigated. This study by Caduff et al. is important as it investigates antigen-specific control of KSHV infection using a humanized mouse model. The authors describe very nicely cytotoxic T cell effector and memory phenotype expansion as well as the viral antigens recognized. Importantly, the authors show that KSHV K6 reactive CD8 T cells are capable of killing KSHV-infected B cells. This finding is significant as it raises the possibility to leverage K6 therapeutically and drive protective T cell responses. Overall, this is an important

manuscript and is well controlled. This will be a study of high importance to the oncogenic virus community.

I only have one minor criticism of the manuscript. The identification of K6 reactive CD8 T cells is interesting and important and should be further discussed. It would be useful to have an expansion of the discussion on K6 genetic diversity in the discussion. Perhaps this genetic diversity is associated with disease progression or treatment outcomes.

Following this reviewer's recommendation we now discuss on page 11 of the revised manuscript the gene copy amplification of the K5-K6 region in KSHV genomes that can be found in Kaposi sarcomas, and cite the respective reference #57. We speculate that increased K6 expression that is associated with less disseminated Kaposi sarcoma could indicate residual immune control even during HIV co-infection, and could suggest K6 as promising T cell antigen to target the respective K6 overexpressing tumors.

Changes in Figures/Supplementary Figures

Figure 1 – NSG-A2 animals are marked as triangles

Figure 2 – NSG-A2 animals are marked as triangles

Figure 3 – NSG-A2 animals are marked as triangles

Figure 4 (new) – T cell depletion during KSHV/EBV infection

Figure 5 – adaption of text font within figure

Supplementary Figure 1 (new) – Histological staining of liver and spleen sections of a KSHV/EBV co-infected animal, providing evidence for EBV and KSHV infection of B cells and KSHV/EBV co-infection; Lymphoma formation in EBV and EBV KSHV infected humanized mice.

Supplementary Figure 2 – former supplementary figure 1

Supplementary Figure 3 (new) – Comparison of T cell expansion and B cell numbers in NSG-A2 and NSG animals

Supplementary Figure 4 (new) – T cell expansion and B cell numbers for mock, KSHV only and KSHV/EBV co-infected animals

Supplementary Figure 5 (new) – dendritic cells and their activation in mock and KSHV/EBV infected animals

Supplementary Figure 6 (new) – Correlation between CD4⁺ T cell activation and CD8⁺ T cell expansion

Supplementary Figure 7 (new) – T cell depletion: EBV loads over time, Lymphoma formation in mock, EBV, KSHV/EBV, EBV T cell depleted and KSHV/EBV T cell depleted animals.

Supplementary Figure 8 – former supplementary figure 2, no changes to the figure content

Supplementary Figure 9 – former supplementary figure 3, no changes to the figure content

Supplementary Figure 10 – (new) – T cell specificity to KSHV peptide pools in KSHV/EBV infected mice

Supplementary Figure 11 – former supplementary figure 4

Supplementary Figure 12 – former supplementary figure 5, no changes to the figure content

Supplementary Figure 13 – former supplementary figure 6, no changes to the figure content

Supplementary Figure 14 – former supplementary figure 7, no changes to the figure content

Supplementary Table S1 – no changes

Supplementary Table S2 – no changes

REVIEWERS' COMMENTS

Reviewer #2 (Remarks to the Author):

No additional suggestions.

Reviewer #3 (Remarks to the Author):

The author has responded to all my comments.

In addition, as a mediator of the response to Reviewer 1, I find the authors to be highly responsive and satisfactory. The additional experimentation and revised figures strengthen the work.